# Multi-centric evaluation of Biomeme Franklin Mobile qPCR for rapid detection of *Mycobacterium ulcerans* in clinical specimens

Michael Frimpong[1,2]*, Venus Nana Boakyewaa Frimpong[2], Hycenth Numfor[3], Valerie Donkeng Donfack[3], Jennifer Seyram Amedior[4], Danielle Emefa Deegbe[4], Baaba Dadson[4], Anthony Ablordey[4], Sara Eyangoh[3], Richard Odame Phillips[2], Sundeep Chaitanya Vedithi[5]

1 Department of Molecular Medicine, School of Medicine and Dentistry, Kwame Nkrumah University of Science and Technology (KNUST), Kumasi, Ghana, 2 Kumasi Centre for Collaborative Research in Tropical Medicine (KCCR), Kwame Nkrumah University of Science and Technology, Kumasi, Ghana, 3 Mycobacteriology Unit, Centre Pasteur du Cameroon (CPC), Yaoundé, Cameroon, 4 Noguchi Memorial Institute of Medical Research (NMIMR), University of Ghana, Accra, Ghana, 5 Department of Biochemistry, University of Cambridge, Cambridge, United Kingdom

* mfrimpong28@gmail.com

**Data Availability Statement:** All relevant data are within the paper and its Supporting Information files.

## Abstract

The gold standard for detection of *Mycobacterium ulcerans* is PCR due to its high accuracy in confirmation of suspected cases. But the available PCR assays are designed for standard size thermocyclers which are immobile and suited for reference laboratories often located long distances from endemic communities. This makes it a challenge to obtain immediate results for patient management. We validated and evaluated a dried reagent-based PCR assay adapted for a handheld, battery-operated, portable thermocycler with the potential to extend diagnostics to endemic communities with limited infrastructure. The diagnostic accuracy of the assay following a multi-center evaluation by three Buruli ulcer reference laboratories with over 300 clinical samples showed sensitivity and specificity of 100–97% and 100–94%, respectively using centralized IS2404 quantitative PCR platform as a reference standard. This assay coupled with a field-friendly extraction method fulfill almost all the target product profiles of Buruli ulcer for decentralized testing at the district, health center and community levels; a key critical action for achieving the NTD Road Map 2030 target for Buruli ulcer.

## Author summary

Early diagnosis of Buruli ulcer remains a major problem in many endemic countries particularly in sub-Saharan Africa which continues to report large wounds as a result. Though reference laboratories with capacity to confirm diagnosis of the disease by recommended PCR method are available in most endemic countries, they are often located long distances from endemic communities and treatment centres. This makes it a challenge to obtain immediate results for patient management. There is therefore the need to have

**Funding:** American Leprosy Missions funded the study with Grant (USA/380) at KCCR that was held by MF. SCV was funded by the American Leprosy Missions grant (RCAM/232) at the University of Cambridge for structure Guided Drug Discovery in Leprosy. The funders had no role in study design, data collection and analysis, decision to publish or preparation of the manuscript.

innovative diagnostic tools that can be fully implemented in local health facilities but not compromising accuracy. We have evaluated a diagnostic tool with accuracy like the recommended method with the potential to be used in health facilities closer to the patients.

## 1. Introduction

Laboratory confirmation of *Mycobacterium ulcerans* infection, Buruli ulcer (BU), a neglected tropical disease reported in over 33 countries worldwide is by PCR [1]. PCR-based techniques developed for diagnosing BU, target the insertion sequence 2404 (IS*2404*) which occurs over 200 times in the genome of *M. ulcerans*. Studies have reported high specificity (100%) and sensitivity (95–98%) for IS*2404* PCR when punch biopsy, swab, and fine needle aspirate (FNA) taken from BU patients were used as diagnostic samples [2,3].

Traditional PCR setup is restricted to reference laboratories located long distances from endemic communities. This situation requires that samples be stored and later transported from endemic communities to reference laboratories for confirmation, which prolongs the time to result of laboratory confirmation as well as increases the amount of money involved in BU case management [4]. This drawback of PCR application in endemic communities has been reported to have resulted in significant reduction (over 40% decrease) in the number of laboratory-confirmed BU cases in endemic countries: Ghana, the Democratic Republic of the Congo, Ivory Coast, Benin and Cameroon in the last 10 years [5]. This has necessitated the WHO to advocate for the development of rapid and simple diagnostic tools with high sensitivity and specificity to facilitate the diagnosis of BU in primary health-care centers close to a patient's residence [6].

The Franklin realtime qPCR machine (Biomeme Inc., USA), a handheld battery-operated thermocycler provides the opportunity of extending molecular diagnosis of infectious diseases to low resources settings. This mobile thermocycler enables the performance of quantitative real-time PCR (qPCR) with a turnaround time of less than one hour compared to other commonly used thermocyclers [7]. The Franklin realtime qPCR machine (Biomeme, USA) works with M1 Sample Prep Cartridge kits and Biomeme Go-Strips (premixed and lyophilised mastermix, primers and probes), shelf-stable and field ready qPCR kits. This extends the application of DNA amplification into the field or into the point-of-care settings for the diagnosis of Buruli ulcer and other Neglected Tropical Diseases (NTDs). Therefore, we sought to validate an established qPCR assay on this platform and evaluate its use for the diagnosis of Buruli ulcer in Ghana and Cameroon.

## 2. Materials and methods

### 2.1. Ethics statement

Approval was given by the Committee for Human Research Publication and Ethics (ref: CHRPE/AP/499/20), School of Medical Sciences, Kwame Nkrumah University of Science and Technology, Ghana and National Ethics Committee for Research in Human Health (ref: 2021/06/1367/CE/CNERSH/SP), Yaounde, Cameroon to use stored samples with no patients' identifiers for this study.

### 2.2. Study design and participants

A multi-country, singled-blinded, diagnostic evaluation study was conducted in three sites: Kumasi Centre for Collaborative Research in Tropical Medicine (KCCR), Kumasi, Ghana;

Centre Pasteur du Cameroon (CPC), Yaoundé, Cameroon; Noguchi Memorial Institute of Medical Research (NMIMR), Accra, Ghana. The formula for comparing two independent proportions for studies comparing sensitivity and specificity of two unpaired design was used to estimate the sample size. A minimum of 100 samples per site were required to achieve 95% confidence [8].

## 2.3. DNA Extraction

Whole genome extraction from serially diluted *Mycobacterium ulcerans* culture isolates $10^7 – 10^\circ$ bact/ml and a panel of fresh clinical samples was carried out at the biosafety level 2 and molecular laboratories at the Kumasi Centre for Collaborative Research into Tropical Medicine, KNUST, Kumasi, Ghana. Swabs and fine-needle aspirates (FNA) were obtained according to the recommended procedure [9] from patients presenting at Ghana Health Service recognized BU treatment centres with clinically suspected BU ulcers and non-ulcerative lesions, respectively. Samples were stored in 700 μl of phosphate buffer saline in 1.5-ml screwed-cap Eppendorf tubes and transported to the laboratory. The M1 Sample Prep Kit (Biomeme, USA) and the Gentra Puregene Cell kit (Qiagen, Germany) were used for extraction.

**2.3.1. DNA extraction with closed cartridge-based DNA extraction system (Biomeme M1 Sample Prep Kits).** Two hundred microliters of sample suspended in buffer were added to BLB (Biomeme Lysis Buffer) and the extraction was performed following manufacturer's instruction. Briefly, using a 1-ml syringe attached with a prep column, the solution (sample + lysis buffer) was drawn up the syringes and pump back out. This was repeated 10 times as indicated. At the 10th pump, the fluid in the syringed prep column was completely expelled into the red section, before the beginning of the next step, without transferring any liquid to the next section. Subsequent wash steps with protein wash (BPW), salt wash (BWB) and dry wash (BDW) were done according to the number of pumps (x times) indicated on the various sections, without transferring any fluid from the previous section to the next until the Air-dry step to eliminate residual fluid. For elution, 500 μL of the eluent was drawn out of the cartridge into 1.5-ml Eppendorf tubes after 5 slow and careful pumps, releasing the purified DNA into the Biomeme elution buffer (BEB).

## 2.4. Quantitative Real-time PCR

**2.4.1. PCR conditions using the Biomeme LyoDNA RT-PCR mastermix.** The Biomeme LyoDNA + IPC (internal positive control) mix (Biomeme Inc. USA) is a blend of freeze-dried deoxyribonucleotide phosphate (dNTP), reaction buffer, magnesium ions and Taq DNA polymerase into which target-specific primers and probes are incorporated. Thus, a prepared working solution of the freeze-dried mix was used in place of the standard qPCR mix reagent (HOT FIREPol mix plus) and tested against target-specific primers and probes, in reference to the standard qPCR protocol [8] on *M. ulcerans* Plasmid, culture isolate (both as templates) and an NTC (No Template Control). PCR runs were done in void stripped PCR tubes at a total reaction volume of 20 μL. Each reaction mix contained 1 μL of 5 μM IS2404 TP2 (5' FAM-CCGTCCAACGCGATCGGCA-BBQ'3), 1 μL each of 10 μM IS2404 TF (5' AAAGCACCA CGCAGCATC T 3'), and IS2404 TR (5' AGCGACCCCAGTGGATTG 3') (TibMolBiol),(10) 4 μL of 2x LyoDNA + IPC mix (Biomeme Inc. USA, 2019) and 11.0 μL Diethyl procarbonate (DEPC) treated water (SolisBioDyne, Estonia) as well as 2 μL of the DNA template. Amplification was done on the Biomeme three9 thermocycler under optimized conditions thermal profile: 95˚C for 60 sec followed by 40 cycles of 95˚C for 1 sec and 60˚ C for 20 sec [10].

**2.4.2. PCR conditions using the Biomeme BU Go-Strips.** The dry reagent based (DRB) BU-Bio PCR reaction mix contained the same reagents and IS2404 primers and probes at the same concentrations as described for the standard IS2404 qPCR assay but with the LyoDNA + IPC mix as earlier stated. The dry bead was prepared into a 3-welled 0.1-ml PCR strips (Biomeme, Inc. USA, 2019) containing freeze dried PCR reagents fused with IS2404 target-specific primers and probes only requiring reconstitution with DNA template for a final reaction volume of 20 μl for amplification. Thermal and cyclic conditions for the BU-Bio Go strip assay was established as: 1 cycle of initial activation at 95˚C for 60 sec followed by 40 cycles of denaturation and annealing/elongation at 95˚C for 1 sec and 60˚ C for 20 sec, respectively for an overall run time of 48 min [7,11,12].

## 2.5. Efficiency, sensitivity, and limit of detection calculations

To assess the efficiency, sensitivity and detection limit of the Biomeme three9 thermocycler and the developed BU-Biomeme (BU-Bio) qPCR assay, a *Mu*-specific DNA standard with known copy numbers for amplified regions, to be used as positive control and calibration template was generated. A 451-bp fragment of IS*2404* sequence covering nucleotides 96540 to 96990 was synthesized by GeneArt Gene Synthesis (Invitrogen, Regensburg, Germany).

Six independent sets of 1:10 serial dilutions of this synthesized quantitative IS*2404* DNA molecular standard (Invitrogen, Regensburg, Germany) were run, each set in triplicates. Testing was done with standard PCR reagents but with reduced durations of cycle steps on the Biomeme three9 thermocycler and compared to standard runs done on the Bio-Rad CFX96 real-time PCR detection system with the same PCR reagents. This was to determine the efficiency of the Biomeme three9 thermocycler, as the least copy number at which amplification occurred and the reproducibility of successful amplifications of the IS*2404* of *M. ulcerans* achieved it was assessed between both detection platforms. Similarly, qPCR runs were done using a quantified *M. ulcerans* genomic DNA from culture isolates in serial dilutions (1 ng-0.1 fg/μL) on both the qPCRs—Biomeme and the Bio-Rad CFX96 real-time PCR system simultaneously.

Identical sets of 1:10 serial dilutions of IS*2404* DNA molecular standards were run simultaneously with the BU-Biomeme qPCR protocol with optimized cyclic conditions on the portable real-time machine in comparison to runs done according to the reference standard IS2404 qPCR assay on the reference in-house real-time detection system, Bio-RadCFX96. This was done to determine the analytical sensitivity of the BU-Biomeme qPCR assay, which employs Biomeme's LyoDNA mix + IPC into which IS2404-specific primers and probe were to be incorporated to make PCR ready dry beads in reference to standard IS2404 qPCR protocol. Co-efficient of determination of reproducibility and detection limits were calculated and extrapolated from a generated standard curve from 1 copy to 10,000,000 copies of Mu plasmid.

## 2.6. Multicentric evaluation of BU Go-Strips assay

This study was conducted at three BU reference laboratories: Kumasi center for Collaborative research (KCCR), Noguchi Medical Research Institute (NMIMR) in Ghana and Center Pasteur DU Cameroon (CPC). The first phase of this study, which involved the development and validation of the BU-Bio PCR assay (analytical sensitivity testing and standard graph generation with *Mu* positive control and calibration template) and the optimization of the Biomeme three9 thermocycler, was done at the KCCR. The second phase, thus, the evaluation of lyophilized BU Go-strip assay and M1 Sample Prep Kits, was done at all 3 reference laboratories. At KCCR however, the diagnostic accuracy (i.e., clinical specificity and sensitivity and positive

and negative predictive values) of the BU PCR Go-Strips assay was determined by testing the assay against 300 clinical samples from 3 study sites.

### 2.7. Statistics

GraphPad Prism v.9 (GraphPad software, San Diego, USA) was used to calculate a semi-log regression of the dataset of repeated amplification runs of qPCR by plotting the mean cycle threshold (CT), against molecules detected of the standard DNA dilutions ($10^6$–$10^\circ$ copies/µl). A probit regression analysis was performed to determine the limit of detection (LOD) in 95% of dilutions for both assays using GraphPad Prism. Descriptive statistics were used to obtain general descriptive information such as median and interquartile ranges from the data. Contingency tables were employed to calculate the sensitivity, specificity and the predictive values in evaluating the assay.

## 3. Results

### 3.1. Optimization of cycle thermal profile /conditions for faster detection of *Mu*

Test runs done with both molecular standard and genomic DNA using stepped-down PCR thermal profile for the Biomeme thermocycler showed improved amplification curves when prior to and after optimization results were analysed as illustrated in (Fig 1). Also, the PCR time was significantly reduced from one 1 hour 34 minutes to 55 minutes. To optimize the reaction volume without compromising assay efficiency, three different qPCR reaction volumes, 15µl, 20µl and 25µl were tested in triplicates with the same concentration of DNA against the optimized conditions for the Biomeme thermocycler to ascertain the reaction volume at which the best amplification results is achieved. Having compared the average quantification cycle values of each reaction volume, the 20µl reaction volume was established to be optimal for the new assay following the reference qPCR protocol as compared to the other volumes.

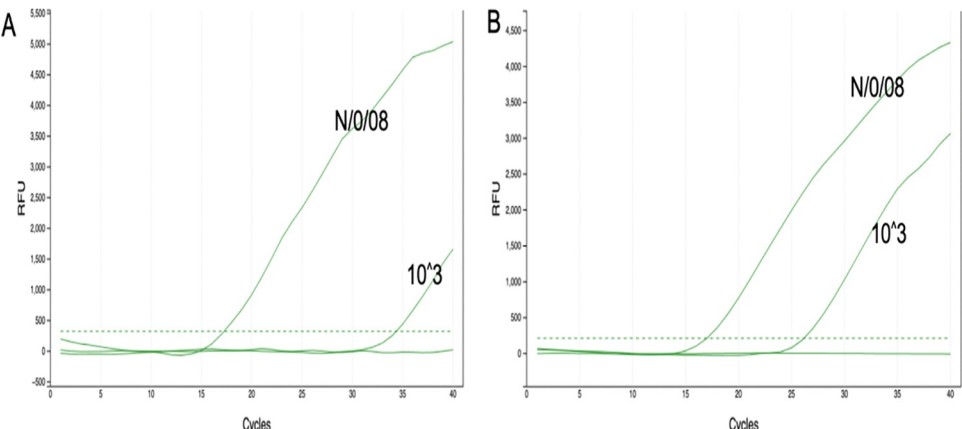

**Fig 1. Representative amplification during optimization, the x-axis is the amplification cycle and the y-axis the relative fluorescent intensity (RFU).** N/0/008 is a sample ID for genome DNA from a known Buruli ulcer positive sample and 10^3 is the concentration of a molecular standard. (A) is the amplification result prior to optimization and (B) is the amplification result of same samples after optimization. The dotted lines are the set fluorescent threshold for a positive call and NTC is the negative template control.

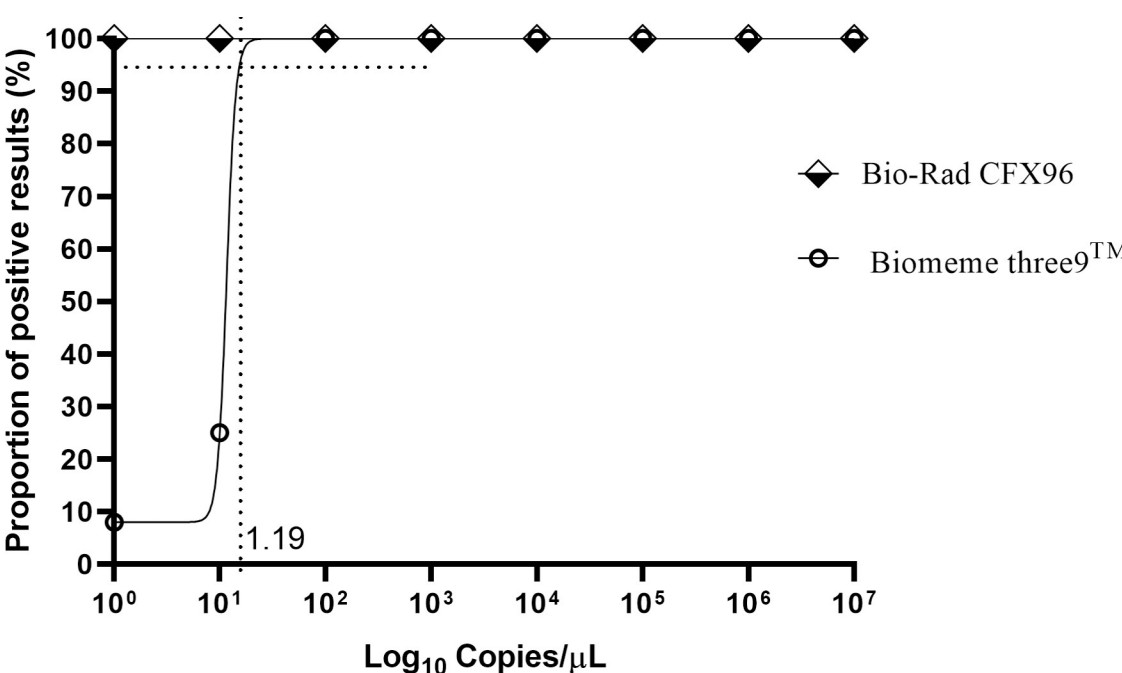

**Fig 2. Probit regression analysis of the detection limit of the assay: The limit of detection in 95% of cases is 15 DNA molecules/ reaction for the assay on Biomeme thermocycler and 1 DNA molecule for the Bio-rad CFX96 thermocycler.**

## 3.2. Comparison of detection rate and limit and analytical of the Biomeme qPCR thermocycler and Bio-Rad CFX96 detection system

The performance of the Biomeme thermocycler with stepped-down durations of thermal conditions was assessed by determining the detection limit of *M. ulcerans* and the reproducibility of successful amplification achieved with the Biomeme thermocycler. Semi-regression and a probit analysis done for a triplicate run of six independent sets of serial dilutions ($10^7$–$10^\circ$ copies/ μL) of molecular standards with a limit of detection at 95% probability was 15 copies for the Biomeme thermocycler as against 1 copy for the reference in-house detection system (Bio-Rad CFX96) (Fig 2).

The BU-Bio qPCR assay is established to be highly specific to only *M. ulcerans* as the real-time qPCR test was carried out using IS2404-specific primers and probe from earlier studies that detect the IS2404 in the *M. ulcerans* genome [13]. Further in silico BLAST analysis of these primers and probe reviewed high specificity for *M. ulcerans*. The analytical sensitivity of the BU-Bio qPCR assay was determined to be highly reproducible as there was a very good correlation of detection between threshold cycles and the copy numbers of DNA molecular standard with equal efficiency ($R^2$ = 0.99) compared to the reference in-house IS2404 qPCR assay ($R^2$ = 0.99) (Fig 3). Both the assays could detect as low as 1 copy ($10^0$) /μL of DNA molecular standard as the percentage proportions of copy numbers detected for each serial dilution from $10^7$ copies down to 1 copy was 100%.

## 3.3. Diagnostic accuracy of IS2404 Dry-reagent-based BU-Biomeme (DRB BU-Bio) assay

The diagnostic accuracy of the IS2404 Dry-reagent-based BU-Biomeme (DRB BU-Bio) assay was determined using a panel of 332 samples from clinically suspected cases of BU for routine diagnosis (Fig 4). These samples were independently tested with the developed assay at KCCR

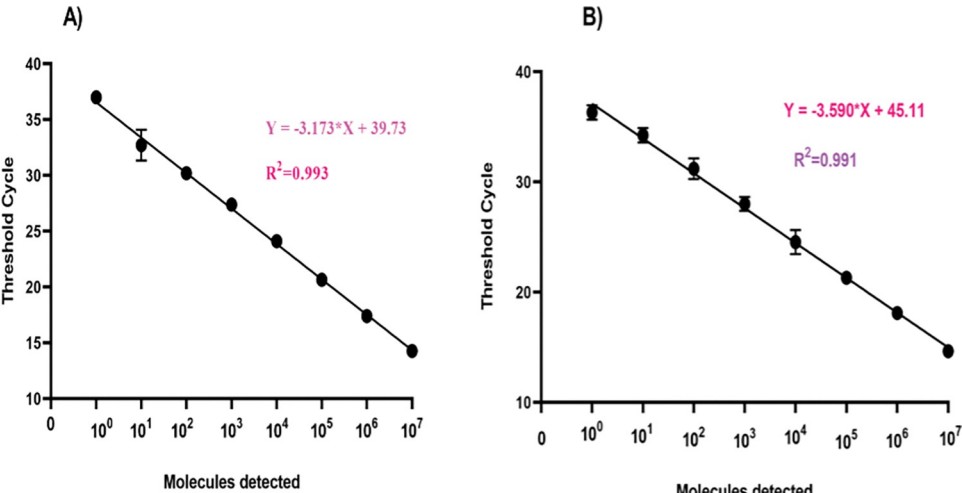

**Fig 3. Reproducibility of the BU-Bio qPCR assay and reference Standard qPCR assay.** Using data set of 6 runs of serial dilutions of *M. ulcerans* molecular standard. The threshold cycle (y-axis) was plotted versus the $\log_{10}$ concentration of standard DNA molecules (x-axis). The error bars represent the range. The BU-Bio qPCR assay (A) had the same detection efficiency $R^2 = 0.993$ as the reference standard assay (B), $R^2 = 0.991$.

(100) and NMIMR (132) in Ghana and CPC (100) in Cameroon. Swab and FNA samples were obtained from patients presenting with ulcers 318 (96%), edema 6 (1.8%), plaques 5 (1.5%) and nodules 3(0.9%). Two hundred and nineteen (219) out of these 332 clinically suspected BU cases were confirmed as BU cases by standard PCR (Table 1). Analysis for the determination of diagnostic accuracy showed an overall sensitivity and specificity of 98% and 96%, respectively. The assay performed extremely well when it was independently evaluated at 3 BU reference laboratories with a sensitivity ranging from 97% to 100% and specificity from 94% to 100% (Table 2).

## 4. Discussion

This study developed, validated, and evaluated the overall test performance of a point-of-need quantitative real-time PCR assay for *M. ulcerans*. Using target-specific primers and probes incorporated into shelf-stable, lyophilized reagents for use with a handheld thermocycler with a simple, rapid field compactible extraction methodology for faster detection of *M. ulcerans*. The PCR assay is the WHO's recommended test for BU case confirmation implemented in several national reference laboratories in BU endemic countries and serves as the reference for new tests [6,14]. But this test in its current form and setup is limited to reference laboratories located long from endemic communities due to its sophistication and complex sample preparation steps [2,5]. A classical characteristic of a point-of-need (PONT) test is its ability to perform rapid detection, therefore the first objective of this study was to optimize the Biomeme Franklin thermocycler to achieve successful amplification under the stepped-down thermal profile for faster detection compared to the standard PCR without compromising the efficiency of the assay. This portable thermocycler (Biomeme, USA) is battery-powered, easy to carry around and can be run off-grid to produce results within 48 minutes. With a mobile app wirelessly connected to the thermocycler, test results are read in real-time or synced into a cloud-based storage for later retrieval. The improved amplification curves obtained post-optimization and the establishment of the 20μl reaction volume as optimal for the new assay following the reference qPCR protocol as compared to the volumes validates the handheld Biomeme thermocycler's capability to perform rapid detection of a target sequence [7].

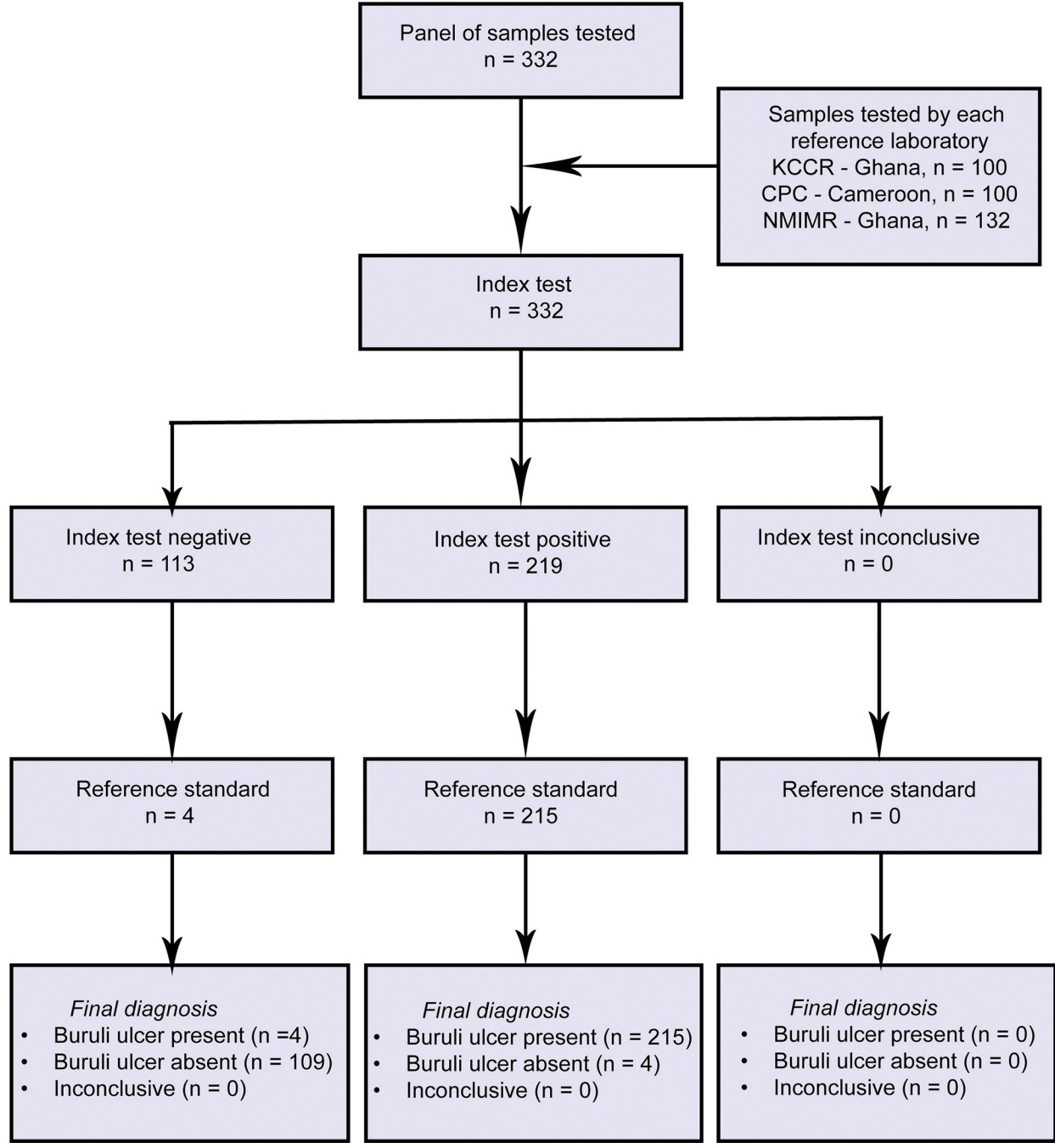

**Fig 4. Flowchart of samples used in determining the diagnostic accuracy of the index test (DRB BU-Bio).** The reference standard refers to the centralized IS2404 qPCR platform.

The performance of the Biomeme thermocycler assessed in this study was compared to the reference test detection system for a 1:10 serially diluted *M. ulcerans* plasmid ($10^7-10°/\mu l$) showed a regression coefficient of determination, $R^2 = 0.78$, compared to that of the Bio-RadCFX96 thermocycler, $R^2 = 0.93$, and a limit of detection at 95% probability, was 15 copies. When a similar assessment done by Nguyen et al, in comparison to the standard bench platform, a regression equation of $R^2$ of 8.0 was estimated for the Biomeme field-portable platform

**Table 1. Demographic and Clinical Characteristics of Study Participants.**

| Characteristics | | No. (%) of total cases (n = 332) |
|---|---|---|
| **Sample type** | Swab | 318 (96) |
| | FNA | 14 (4) |
| **Presented lesions** | Nodule | 3 (0.9) |
| | Oedema | 6 (1.8) |
| | Plaque | 5 (1.5) |
| | Ulcer | 318 (95.8) |
| **Category of Lesion** | I | 36 (11.0) |
| | II | 35 (10) |
| | III | 29 (9.0) |
| | N/I | 232 (70.0) |
| **Confirmed cases** n | | 219 (66) |

IQR, Interquartile range; I- single lesions ≤ 5cm in diameter; II- single lesions 5-15cm in diameter; III- single lesions ≥ 15cm in diameter, multiple lesions, lesion at critical sites (e.g., eyes and genitals), osteomyelitis; N/I–not indicated; n–total number.

[11]. This indicates that detection efficiency of a portable PCR is not as high as the stationery bigger PCR systems but good enough for detection of pathogens in resource limited settings.

The availability of the particularly sensitive and specific IS*2404* PCR assays for the detection of *M. ulcerans* has raised the criterion for an optimal rapid test as a PONT for Buruli ulcer even at the primary healthcare level. According to the target product profile (TPP), despite the urgent necessity, the newly developed test must be not only rapid but as sensitive and specific as standard PCR [6]. The analytical sensitivity of the BU-Biomeme qPCR assay developed in this study was equal to that of the standard IS2404 qPCR assay. Both assays could detect as low as 1 copy/µl of DNA molecular standard with percentage proportions of copy numbers detected for each serial dilution from $10^7$ copies down to 1 copy being 100%. There was a very good correlation of detection between threshold cycles and the copy numbers of DNA molecular standard with equal efficiency, $R^2 = 0.99$ in both assays. Based on these results it was demonstrable that the portable qPCR assay was reproducible and highly sensitive as the reference standard. These results are similar to that demonstrated by Frimpong M, et al where

**Table 2. Clinical sensitivity and specificity of BU-Bioassay compared to standard qPCR assay.**

| Study sites | | DRB BU-BIO qPCR | | | | N | DRB BU-BIO qPCR | Result tables | |
|---|---|---|---|---|---|---|---|---|---|
| | | | | | | | | Reference Test | |
| | | Sensitivity | Specificity | PPV | NPV | | | Real-time RT-PCR | |
| | | | | | | | | Pos | neg |
| KCCR | Estimate: | 1 | 1 | 1 | 1 | 100 | Pos | 57 | 0 |
| | 95%CI: | [0.94;1.0] | [0.92;1.0] | [0.94;1.0] | [0.92;1.0] | | Neg | 0 | 43 |
| CPC | Estimate: | 0.98 | 0.94 | 0.94 | 0.98 | 100 | Pos | 51 | 3 |
| | 95%CI: | [0.90;0.99] | [0.83;0.99] | [0.85;0.98] | [0.87;0.99] | | Neg | 1 | 45 |
| NMIMR | Estimate: | 0.97 | 0.95 | 0.99 | 0.87 | 132 | Pos | 107 | 1 |
| | 95%CI: | [0.92;0.99] | [0.78;0.99] | [0.95;1.0] | [0.69;0.96] | | Neg | 3 | 21 |
| Total | Estimate: | 0.98 | 0.96 | 0.98 | 0.96 | 332 | Pos | 215 | 4 |
| | 95%CI: | [0.95;0.99] | [0.91;0.99] | [0.95;0.99] | [0.91;0.99] | | Neg | 4 | 109 |

pos (+): Positive, neg (-): Negative, PPV: Positive Predictive Value, NPV: Negative Predictive Value

semi-regression and probit analyses were done for the reference qPCR assay for *M. ulcerans* showed $R^2$ = 0.99 and a detection limit of 1 copy, juxtaposed to that of RPA, $R^2$ = 0.92 and a limit of 45 copies [15].

Template preparation procedure and nucleic acid recovery or yield from fresh swab and FNA samples using the Biomeme M1 Sample Prep kit (Biomeme, USA) were assessed against the Gentra Puregene DNA Isolation kit (Qiagen, Germany) currently used in the reference laboratory. The M1 Sample Prep kit, a filtration-based equipment-free method allows for template preparation within 2–5 minutes. The GenoLyse Kit (Hain Life Science, Germany) extraction employed in this study was according to the BU LABNET protocol, a harmonization process towards a centralized sample preparation for laboratory confirmation for BU [16], hence an alternative comparator. Having done a comparative analysis, using the quantification cycle Cq values as a measure of DNA yield obtained from all three procedures, for each sample, there was no significant difference between the DNA yields (P = 0.952) as also demonstrated by Hole, K., & Nfon, C. and Tomaszewicz Brown A, et al. where although potentially false negatives due to low viral load or lower RNA recovery were observed with Biomeme, there was no statistically significant difference reached [11,12]. In addition to a field-compatible simple, rapid and equipment-free extraction procedure, a dry reagent-based qPCR assay overcomes the laborious and technical challenges that accompany the implementation of PCR procedures for diagnosis in tropical countries. The diagnostic accuracy of the BU-Biomeme Go-Strip assay (DRB BU-Bio) assay independently evaluated with over 300 clinical samples revealed sensitivity and specificity of 100% - 97% and 100% - 94%, respectively, across three reference laboratories. This was highly promising for a point-of-need test, and some studies have supported the assumption that sensitivity and specificity of an IS*2404* developed PCR assay could approach 100% under favourable conditions [13].

The test procedure for this assay in addition to its performance as enumerated above makes it an ideal rapid test for diagnosis of Buruli ulcer at the primary healthcare level since it meets most of the parameters of target product profile [17]. End users can be trained within a day to fully conduct the test, there is less sample preparation and operator steps with an easy interpretation of results through a mobile App connected to the portable thermocycler via Bluetooth without the need for internet connection. An important operational characteristic for the assay is the no cold chain requirement for reagents making it well-suited for use in resource-limited settings. The study was conducted with well characterized stored samples, and this has a limitation for the assessment of the real diagnostic performance. We recommended a prospective study to evaluate this tool at the point of patient care to determine the true diagnostic performance as a point of need test. Also the cost per test, which is currently between $20 and 25, mainly due to the cost of extraction kit is a potential limitation for implementation. An alternative less expensive extraction kit could be used instead.

In conclusion, we developed a DRB BU-BIO assay that has been independently evaluated in three reference laboratories and is now commercially available. This assay meets most of the characteristics in the target product profile (TPP) for a rapid test for diagnosis of BU at the primary Healthcare facility level. This is a major step in taking the gold standard test to the point of need to aid early detection and reduce the challenges of obtaining immediate results for patient management.

## Supporting information

**S1 STARD Checklist. STARD checklist for reporting diagnostic accuracy of the Biomeme Franklin mobile qPCR platform for diagnosis of Mycobacterium ulceranns infection.** (PDF)

**S1 Raw data. Raw data of sample characteristics and results for index and reference standard.**
(XLSX)

## Author Contributions

**Conceptualization:** Michael Frimpong, Anthony Ablordey, Sara Eyangoh, Richard Odame Phillips, Sundeep Chaitanya Vedithi.

**Data curation:** Venus Nana Boakyewaa Frimpong, Hycenth Numfor, Valerie Donkeng Donfack, Jennifer Seyram Amedior, Danielle Emefa Deegbe, Baaba Dadson.

**Formal analysis:** Michael Frimpong, Venus Nana Boakyewaa Frimpong.

**Funding acquisition:** Michael Frimpong, Sundeep Chaitanya Vedithi.

**Investigation:** Michael Frimpong, Venus Nana Boakyewaa Frimpong, Hycenth Numfor, Valerie Donkeng Donfack, Jennifer Seyram Amedior, Danielle Emefa Deegbe, Baaba Dadson.

**Methodology:** Michael Frimpong, Venus Nana Boakyewaa Frimpong, Anthony Ablordey, Sara Eyangoh, Richard Odame Phillips.

**Project administration:** Michael Frimpong.

**Resources:** Anthony Ablordey, Sara Eyangoh, Richard Odame Phillips, Sundeep Chaitanya Vedithi.

**Supervision:** Michael Frimpong, Anthony Ablordey, Sara Eyangoh, Richard Odame Phillips.

**Validation:** Michael Frimpong, Anthony Ablordey, Sara Eyangoh, Richard Odame Phillips, Sundeep Chaitanya Vedithi.

**Writing – original draft:** Michael Frimpong, Venus Nana Boakyewaa Frimpong.

**Writing – review & editing:** Michael Frimpong, Hycenth Numfor, Valerie Donkeng Donfack, Jennifer Seyram Amedior, Danielle Emefa Deegbe, Baaba Dadson, Anthony Ablordey, Sara Eyangoh, Richard Odame Phillips, Sundeep Chaitanya Vedithi.

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
