## [Decision Letter · Decision Letter 0]

12 Mar 2023

Dear Dr Frimpong,

Thank you very much for submitting your manuscript "Multi-centric evaluation of Biomeme Franklin Mobile qPCR for rapid detection of Mycobacterium ulcerans in clinical specimens" for consideration at PLOS Neglected Tropical Diseases. As with all papers reviewed by the journal, your manuscript was reviewed by members of the editorial board and by several independent reviewers. In light of the reviews (below this email), we would like to invite the resubmission of a significantly-revised version that takes into account the reviewers' comments. 

We cannot make any decision about publication until we have seen the revised manuscript and your response to the reviewers' comments. Your revised manuscript is also likely to be sent to reviewers for further evaluation.

Sincerely,

Joseph Vinetz

Section Editor

Reviewer's Responses to Questions

**Key Review Criteria Required for Acceptance?**

**Methods**

-Are the objectives of the study clearly articulated with a clear testable hypothesis stated?

-Is the study design appropriate to address the stated objectives?

-Is the population clearly described and appropriate for the hypothesis being tested?

-Is the sample size sufficient to ensure adequate power to address the hypothesis being tested?

-Were correct statistical analysis used to support conclusions?

-Are there concerns about ethical or regulatory requirements being met?

Reviewer #1: Yes to all, but the population. See comments below

**Results**

-Does the analysis presented match the analysis plan?

-Are the results clearly and completely presented?

-Are the figures (Tables, Images) of sufficient quality for clarity?

Reviewer #1: yes

**Conclusions**

-Are the conclusions supported by the data presented?

-Are the limitations of analysis clearly described?

-Do the authors discuss how these data can be helpful to advance our understanding of the topic under study?

-Is public health relevance addressed?

Reviewer #1: yes, partially

**Editorial and Data Presentation Modifications?**

Reviewer #1: (No Response)

**Summary and General Comments**

Reviewer #1: The present work deals with the presentation of a miniaturized real-time PCR system plus DNA extraction method and its comparison to conventional qPCR method used in the diagnosis of Buruli ulcer (BU). PCR is the method recommended by the World Health Organization (WHO) to confirm cases of BU. In principle this miniaturized PCR would allow decentralized diagnosis enabling early treatment, and ths contributing to WHO goals.

The work is relevant, but I believe there are some aspects n the manuscript that would need to be addressed:

1- In the abstract and introduction the authors indicate that current PCR methods are not as able to be implemented in decentalized settings as it would be this miniaturized PCR. However, data on usability (DNA extraction and PCR methods) are not presented. This makes difficult to see the potential advantages of this new PCR system. This is also part of the discussion, but data or work addressing usability is missed.

2- Abstract. Indicate sensitivity and specificity against which reference test.

3- Introduction, lines 59-60. Indicate reference test(s) in those studies.

4- Materials and Methods. The manuscript would benefit of adherence to STARD guidelines. A STARD workflow would be very helpful to understand the different processes.

5- Materials and Methods, line 90. what is the author definition for a Phase II diagnostic evaluation study?

6- Materials and Methods, lines 93-96, please provide the formula or a reference.

7- Note that the study was conducted with well characterized stored samples, and this is a limitation for the assessment of the real diagnostic performance.

8- Materials and Methods, line 115. 200 uL sample or sample eluted/diluted in a buffer, what would be the proxy to understand the real amount of clinical sample used? 

9- Materials and Methods, line 125-126. Indicate elution volume

10- Materials and Methods. Did the authors used the same amount of DNA in the conventional real-time PCR and the strips-based real-time PCR? this is not clear and is needed for proper comparison.

11- Results, line 275. As the study samples were characterized previously by PCR, this increases pre test probability and overestimates sensitivity and my underestimate specificity. Relate this to my comment 7.

12- Discussion, lines 314-315. THis was done in a panel of well-characterized stored clinical samples. This has limitations, see comments 7 and 11.

13- Discussion, lines 339-340. As it seems developers of the method (adapted to the Biomeme system) were involved in this evaluation, I don't think it can said that evaluation was independent.

PLOS authors have the option to publish the peer review history of their article (what does this mean?). If published, this will include your full peer review and any attached files.

Reviewer #1: No
---

## [Decision Letter · Decision Letter 1]

11 May 2023

Dear Dr Frimpong,

We are pleased to inform you that your manuscript 'Multi-centric evaluation of Biomeme Franklin Mobile qPCR for rapid detection of Mycobacterium ulcerans in clinical specimens' has been provisionally accepted for publication in PLOS Neglected Tropical Diseases.

Best regards,

Joseph M. Vinetz

Section Editor

Joseph Vinetz

Section Editor

Reviewer's Responses to Questions

**Key Review Criteria Required for Acceptance?**

**Methods**

-Are the objectives of the study clearly articulated with a clear testable hypothesis stated?

-Is the study design appropriate to address the stated objectives?

-Is the population clearly described and appropriate for the hypothesis being tested?

-Is the sample size sufficient to ensure adequate power to address the hypothesis being tested?

-Were correct statistical analysis used to support conclusions?

-Are there concerns about ethical or regulatory requirements being met?

Reviewer #1: Yes to all

**Results**

-Does the analysis presented match the analysis plan?

-Are the results clearly and completely presented?

-Are the figures (Tables, Images) of sufficient quality for clarity?

Reviewer #1: Yes to all

**Conclusions**

-Are the conclusions supported by the data presented?

-Are the limitations of analysis clearly described?

-Do the authors discuss how these data can be helpful to advance our understanding of the topic under study?

-Is public health relevance addressed?

Reviewer #1: Yes to all

**Editorial and Data Presentation Modifications?**

Reviewer #1: (No Response)

**Summary and General Comments**

Reviewer #1: The authors addressed my comments from teh previous review. The current manuscript addresses the changes made. So I would not require further action, but:

1. reference 8 is incomplete

2. line numbers in response to reviewrs do not correspond to line numbers in the new manuscript

And:

I don't think answers to comments 11 and 12 are right, but the changes made in the Discusion section as per the answr to comment 7 are OK regarding comments 11 and 12 too.

PLOS authors have the option to publish the peer review history of their article (what does this mean?). If published, this will include your full peer review and any attached files.

Reviewer #1: No

---

## [Editor Report · Acceptance letter]

22 May 2023

Dear Dr Frimpong,

We are delighted to inform you that your manuscript, "Multi-centric evaluation of Biomeme Franklin Mobile qPCR for rapid detection of Mycobacterium ulcerans in clinical specimens," has been formally accepted for publication in PLOS Neglected Tropical Diseases.

Best regards,

Shaden Kamhawi

co-Editor-in-Chief

Paul Brindley

co-Editor-in-Chief
